

# A rapid method for methanol quantification in spirits using UV-visible spectroscopy and FTIR

Ronick S. Shadrack[1], Krishna K. Kotra[2], Daniel Tari[1], Hancy Tabi[1], Jacinta Botleng[1], Rolina Kelep[1] and Ladyshia Regenvanu[1]

[1] Laboratory, Vanuatu Bureau of Standards, Port Vila, Vanuatu
[2] School of Agriculture, Geography, Environment, Ocean and Natural Sciences (SAGEONS), The University of the South Pacific, Emalus Campus, Port Vila, Shefa, Vanuatu

Corresponding author
Ronick S. Shadrack,
rspenly@gmail.com

## ABSTRACT

Although standards methods of food safety assessment are important, these methods are relatively expensive and require intensive work and time. In alcohol beverage industries, ultraviolet visible (UV-Vis) spectroscopy and Fourier transform infrared (FTIR) spectroscopy are reliable techniques for quality assessment of alcohol, however, testing methods are often varying with calibration techniques and instrument specification. In this work, methanol content in ethanol was assessed in two approaches using UV-Vis with a developed calibration technique and FTIR spectroscopy with a factory default scan function at every 2 nanometer (nm). For UV-Vis method, potassium dichromate was used as the chromogenic reagent, tested with methanol concentration ranging from 0.12% to 1% ($mV^{-1}$). For FTIR method, spectra data was collected every 2 nm interval and calibration curve was built by increasing methanol ratio from 0% to 40% ($mV^{-1}$) at the expense of ethanol while keeping deionised (DO) water constant at 5% ($mV^{-1}$) concentration. This helps gauge the change in methanol concentration relative to ethanol. Results of analysis using UV-Vis showed a strong negative correlation for methanol concentration and absorbance value at UV region from 900 to 1,100 $cm^{-1}$ (r = 98.00, RMSE = 0.023) relative to increasing ethanol concentration. A strong peak was observed for methanol concentration at spectral region of 975 $cm^{-1}$ which is related to the methanoic acid C-O bond. The FTIR spectra region at 900 to 1,050 $cm^{-1}$ was used for observing methanol concentration with absorbance. A strong correlation was established from spectral region of 1,010 to 1,026 $cm^{-1}$, enabling quantification of methanol (r = 0.99, RMSEC = 0.55). Methanol peak was observed at 1,020 $cm^{-1}$ region of the spectrum. A set of experimental repetition was made with methanol concentration of 0.02% to 0.5% and 0.1% to 5% for UV-Vis and FTIR, respectively, to determine the limit of detection (LOD) and limit of quantification (LOQ). The observation showed a 0.04% and 0.29% ($mV^{-1}$) LOD for UV-Vis and FTIR method, respectively. The LOQ was 0.12% and 0.89% ($mV^{-1}$) for UV-Vis and FTIR respectively. The integration of UV-Vis with potassium dichromate as chromogenic reagent and FTIR spectroscopy with comparatively 50% less data point still present a significant advancement in the test method for safety and quality control of alcohol beverage products. These techniques not only enhance the ability to detect harmful substances but also provide a cost-effective and rapid alternative to traditional

methods, making them invaluable tools for distilleries aiming to uphold high standards of quality.

# INTRODUCTION

Techniques have been exploited to determine methanol and ethanol in alcoholic beverages. Several methods include gas chromatography mass spectrometry (GC-MS) as reference method (*Debebe et al., 2017*; *Tadesse et al., 2017*), density and refractive measurements, spectroscopic technique using Schiff's reagent or chromatropic acid, gas chromatography and Fourier transform infrared spectroscopy (GC-FTIR) (*Sharma, Sharma & Lahiri, 2009*), Fourier transform infrared spectroscopy (*Coldea et al., 2013*), Fourier transform mid-infrared spectrophotometry (FT-MIR) and near-infrared spectrophotometry (NIR) (*Debebe et al., 2017*), and ultraviolet near-infrared spectroscopy (*Tadesse et al., 2017*). Infrared (IR) spectroscopy and UV-Vis are two well established tools in analytical chemistry. Both instrument offers non-invasive and non-destructive method of analysis that is rapid and applicable to wide range of sample types (*Khalid, Ishak & Chowdhury, 2024*; *Johnson et al., 2023*). Historically, UV-Vis and infrared spectroscopy have been utilised extensively due to low price of instruments, low operating cost and short turnaround time for results. Although IR spectroscopy has some advantages over UV-Vis in terms of wide range of sample types, they are both important in food industry for quality control measures (*Sharma, Sharma & Lahiri, 2009*). Some challenges with IR spectroscopy includes interpretation of spectral data from complex mixtures, and the need to develop and maintain robust quantitative calibration model (*Bureau, Cozzolino & Clark, 2019*). The use of UV-Vis and FTIR spectroscopy in quantification of higher alcohol is often neglected due to low performance in terms of accuracy, limit of detection (LOD), limit of quantification (LOQ) and shifting region of interferogram due to thermal and structural properties of alcohol mixtures compared to recognised GC-MS method (*Santos et al., 2023*). Additionally, in most previous studies, the factory default function of FTIR was programmed to collect spectral data at every 1 nanometre (nm) interval (*Coldea et al., 2013*; *Santos et al., 2023*; *Thanasi et al., 2024*). The knowledge on the potential use of FTIR with a 2 nm interval spectral scan data in quality control of high alcohol in alcohol beverage is rare.

The use of FTIR and UV-Vis in developing countries still remains high in the food monitoring sector despite the limitations (*Fakayode et al., 2020*). Previous studies have reported the use of potassium dichromate as chromatographic indicator which oxidises alcohol for calibration of UV-Vis and the quantification of methanol in biodiesel washing wastewater (*Santos et al., 2023*; *Conceição et al., 2018*; *Pérez-Ponce & de La Guardia, 1998*). The present study intends to investigate the use of potassium dichromate ($K_2Cr_2O_7$) as indicator for quantification of methanol in distilled spirit using UV-Vis method. This

study will also investigate the potential of quantifying methanol in distilled spirit using an FTIR of 2 nm interval default scan function. The calibration model will be developed to account for spectral shifts and overlapping spectral regions in alcohol mixtures, caused by variation in their structural characteristics and thermal properties. The model will integrate these factors to enhance the accuracy of spectral analysis, ensuring reliable differentiation and quantification of the components within the mixture.

In the Pacific Island countries including Vanuatu, distilled spirit market is relatively new and the need for advance technology with higher operating cost such as GC-MS for quantification of methanol in spirits is not practical, though necessary. It has been confirmed that alcohol is a leading risk factor for disease and injury in the Pacific Island countries (*Whiteford et al., 2013*; *Kessaram et al., 2016*). Thus, there is a need to investigate a simple and rapid method for high alcohol quantification of methanol in distilled spirit to meet the safe limit recommended by International food regulatory organisations such as International Standards Organisation (ISO), U.S. Food and Drug Administration (FDA) and European Food Association. In the European countries the maximum acceptance limit for methanol in distilled spirit is 10 g methanol/L of alcohol (*Paine & Dayan, 2001*) while in the US, methanol limit in alcohol is 7 g methanol/L alcohol (*Botelho et al., 2020*). This study relies on two mentioned standards for quality control of methanol in distilled spirits irrespective of accurate estimation of the amount that are well below instrumental LOD. The precision and accuracy of FTIR spectra scan at every two nanometres will also be investigated. Portions of this text from this section and subsequent section were previously published as part of a preprint (*Shadrack et al., 2024*).

## METHODS

### Materials

Reagents used in the current analysis are methanol ($CH_3OH$, Merck, HPLC, ≥99.9%), sulfuric acid ($H_2SO_4$, Merck, 97%), potassium dichromate ($K_2Cr_2O_7$, Cinética, 99%), ethanol ($CH_3CH_2OH$, Qhemis, 99.5%), and DO water. All reagents were used as received. The distilled spirits tested in this study was donated by the 83 island distillery in Port Vila, Vanuatu. The tested sample were distilled from fermented sugarcane and were at heads stages of purification for high alcohol.

### Detection and quantification of methanol by FTIR

Prior to collection of the calibration spectra, room environment was controlled with temperature and humidity being 21 °C and 35%, respectively. For the methanol FTIR calibration model, DO water volume was held constant at 60% ($mV^{-1}$) of the calibration solution while methanol concentration increases from 1.8% to 40% ($mV^{-1}$) at the expense of ethanol in 21 data points. The blank was made of deionised water only. The mid IR spectra by Fourier transform were obtained by a Bruker spectrometer, model Alpha II, coupled with ATR single-reflectance cell, and diamond prism with a contact diameter of 1.8 mm. The FTIR instrument used was pre-calibrated to obtained spectra point at every two nanometres which equate to half of the total data for FTIR instrument that obtained data in

every nanometre. Spectra were obtained in the spectral region between 500 to 4,000 cm$^{-1}$ with 23 scans in absorbance mode, resolution of 4 cm$^{-1}$, apodization by the Sqr Triangle function, and 20 µl of each sample entry was used. These operating conditions led to 20 s analysis time. Area under peaks were used as analytical signal to construct the calibration model.

## Detection and quantification of methanol by UV-Vis

The absorbance read at 900 to 1,100 cm$^{-1}$ were obtained using diffuse reflectance spectroscopy in the visible region (325–1,100 cm$^{-1}$) in a Shimadzu Spectrophotometer, model UV—1800. To construct the standard curve, standard preparation was described as follows; methanol volume was held constant at 5% (mV$^{-1}$) of the standard solution while ethanol concentration increases from 5% to 40% (mV$^{-1}$) at the expense of DO water in 16 data points. Before measurement in the spectrophotometer, the standards for constructing the calibration curve was prepared using the following steps; 5 mL of $H_2SO_4$ was transferred into each diluted samples followed by addition of 1 mL of $K_2Cr_2O_7$ solution (10% mV$^{-1}$), and, finally, the solutions were mixed for approximately 2 min by vortexing. The finished mixtures were analysed using a blank sample prepared in the same way as described above by transferring $H_2SO_4$ and the $K_2Cr_2O_7$ solution into 5 mL of D0 water without adding methanol. The LOD and LOQ for the two method was determined with spiked methanol concentration of 0.02% to 0.5% and 0.1% to 5% for UV-Vis and FTIR, respectively.

## Statistical analysis model

Based on the FTIR and UV-Vis spectra, the analysis were performed at frequency intensities recorded in the domain 1,010–1,026 cm$^{-1}$ and 900–1,100 cm$^{-1}$, respectively. Principle component analysis (PCA) was performed by Unscrambler 10.4 Software, version 10.4 (Camo Software AS, Oslo, Norway) for sample variability and discrimination. The peak areas from the specified wave numbers were analysed for their concentration and prediction using partial least squares (PLS) regression with the same software. For the model development, 70% of sample data was used in learning set and the remaining 30% for the validation set. Spectra were cantered prior to further analysis. The PLS model was pre-weighted and a full cross validation was performed. A general linear regression model (GLM) was used to evaluate the limit of detection (LOD) and limit of quantification (LOQ) for the predictive model.

# RESULTS AND DISCUSSION

## Identification of methanol and ethanol in the UV-Vis spectra of pure standards and a spiked sample

In order to distinguish the colour differences in the pure methanol and ethanol standard under oxidation reaction with acid dichromate, a trial was made with the sub sample of known concentration following the method. The standard methanol (99.9%) (Fig. 1A) and ethanol (99.8%) (Fig. 1B) in the chemical mixtures and the blank (DO water) (Fig. 1D) can be easily differentiated with the colour differences. After the oxidation of methanol, the

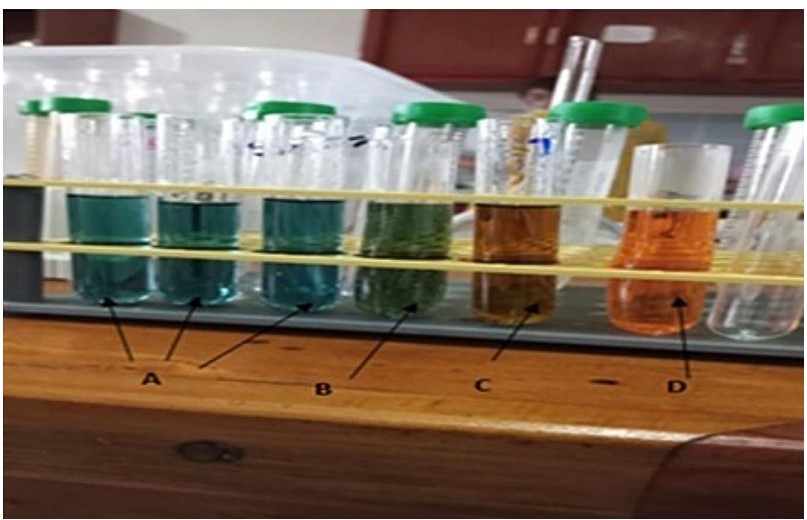

**Figure 1 Methanol, ethanol and spiked methanol in acid dichromate reagent with color differences.**
(A) Methanol (99.8%) in acid dichromate, (B) ethanol (99.6%) in acid dichromate, (C) spiked methanol and ethanol (1:2% mV$^{-1}$) in DO, with acid dichromate and, (D) reagent black in acid dichromate.

samples were revealed a colour scale of golden yellow for blank sample (Fig. 1D) and a golden teal (Fig. 1A) for methanol. In the presence of ethanol, methanol presence by colour is challenging to differentiate due to presence of golden green colour produced from oxidation of ethanol by dichromate (Fig. 1C).

Collection of sample spectrum from a full spectrum scan is a common technique to determine peaks absorption of molecule in any given sample (*Mohamed et al., 2017*). In this experiment, in order to identify peaks relative to change in concentration of methanol and ethanol in acid dichromate, the whole sample spectrum was scanned at region of 325 to 1,100 cm$^{-1}$ (Fig. 2). Peak intensity change in ethanol was observed at 600 nm while methanol change in peak intensity was strongly correlated at 975 cm$^{-1}$. As the concentration of methanol decrease relative to ethanol, peak intensity reflecting the change was observed in the 975 cm$^{-1}$ region of the spectrum. The reaction and the change in oxidation of methanol in acidic condition and the related by-product is shown in the Eq. (1) below, according to *Santos et al. (2023)*.

$$CH_3OH_{(l)} + H_2Cr_2O_{7(aq)} + 4H_2SO_{4(l)} \rightarrow CO_{2(g)} + 6H_2O_{(l)}$$
$$+ Cr_2(SO_4)_{3(aq)} + K_2SO_{4(aq)}. \tag{1}$$

After the methanol oxidation process, the samples were revealed with colours ranging from a scale of golden yellow to green (Fig. 1A). As shown in Eq. (1), it is suggested that methanol was wholly oxidized to $CO_2$ and $H_2O$ in the presence of $K_2Cr_2O_7$ under a strongly acidic medium (*Rao, 1966*). At the same time that $Cr^{6+}$ (golden yellow) was reduced to $Cr^{3+}$ (teal colour). It is also believed that this reduction may pass through an intermediate state of $Cr^{4+}$ due to the reddish-brown coloration (*Rao, 1966*). The intensity

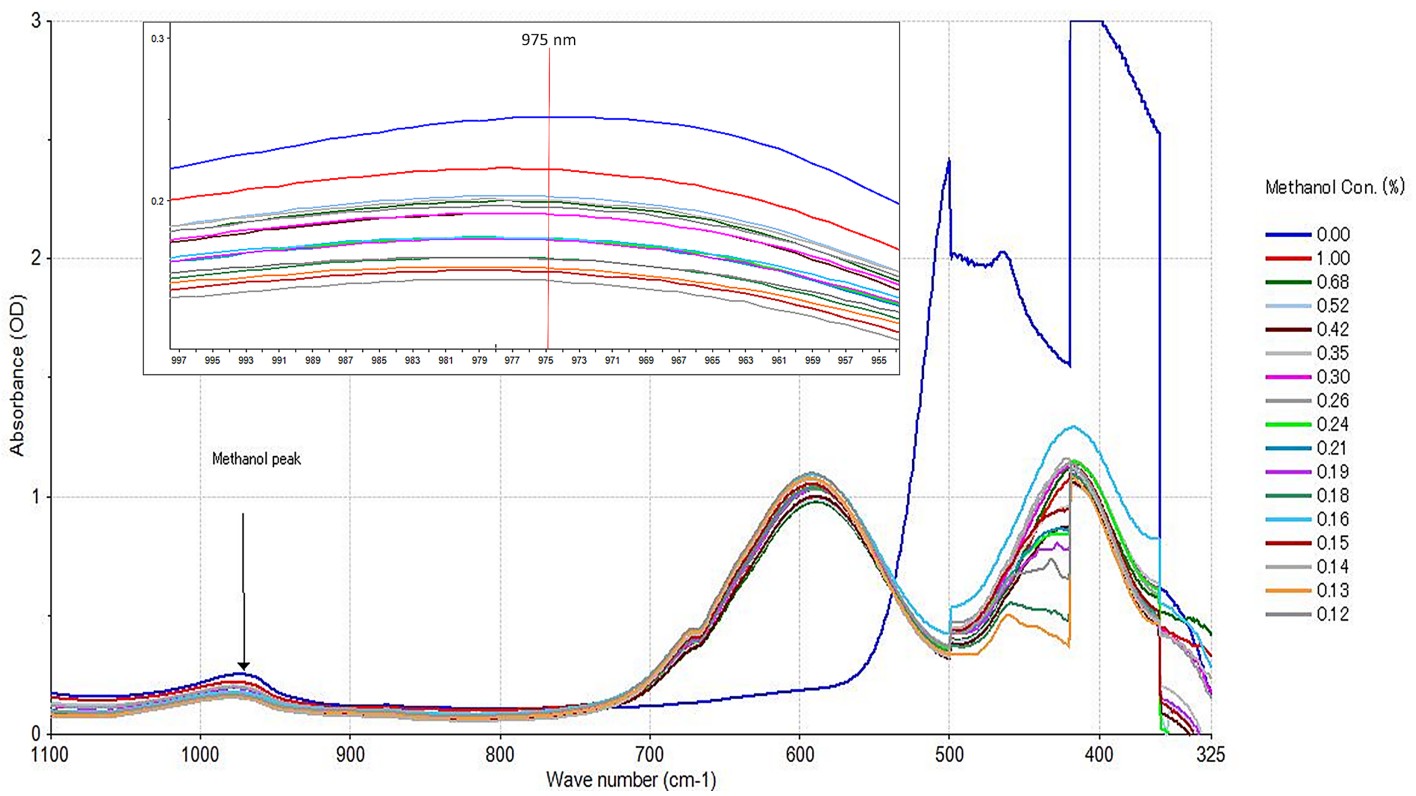

**Figure 2 Spectral distribution of spiked methanol in ethanol used in UV-Vis predictive model development.** UV-Vis spectra of spike methanol using the acid dichromate method. Ethanol peaks at 600 nm, and methanol peaks at 975 nm. The 975 nm methanol peak is shown with a red line on the zoom version.

of the colour read at 600 cm$^{-1}$ was inversely proportional to the concentration of methanol in each sample observed at 975 cm$^{-1}$ of the spectrum. This result is in contrast to *Santos et al. (2023)* who reported that methanol detection with UV-Vis in biodiesel waste water was observed at 600 cm$^{-1}$. Here, colour intensity at 600 cm$^{-1}$ was propositional to ethanol concentration.

## Calibration and validation of PLS procedure for methanol

In view of the UV-Vis spectra display in Fig. 2, there was a gradual decrease in absorbance as the concentration of methanol decrease relative to increasing ethanol concentration. Such behaviour was reported by *Yuan et al. (2022)*. The PLS model development from the acid dichromate UV-Vis method showed a good correlation with methanol concentration at 975 cm$^{-1}$ spectra region with a correlation coefficient of r = 0.99 and root mean square error coefficient (RMSEC) of 0.007 (Fig. 3). The validation of the model revealed a strong correlation of r = 0.96, with RMSECV of 0.028% (mV$^{-1}$) (Fig. 4). This study has confirmed the UV-Vis method using acid dichromate as potential chromatic agent for methanol quantification in alcohol beverages especially distilled spirits. The use of PLS model showed a good predictive model and such technique is cost effective for small businesses

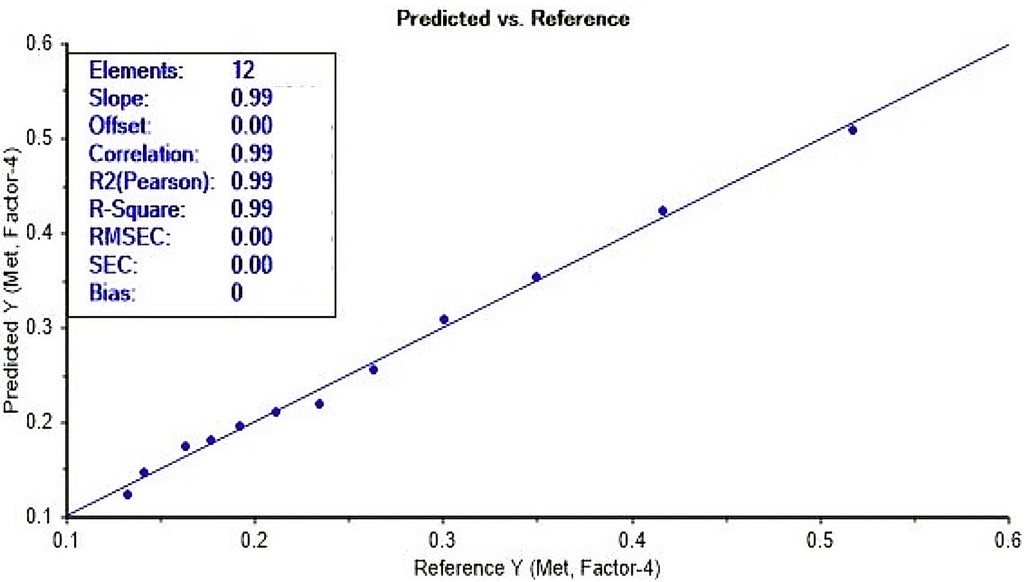

**Figure 3 Calibration curve for methanol prediction using UV-Vis with acid dichromate.** Calibration set for methanol prediction model with UV-Vis (900 to 1,100 cm$^{-1}$) using acid dichromate.

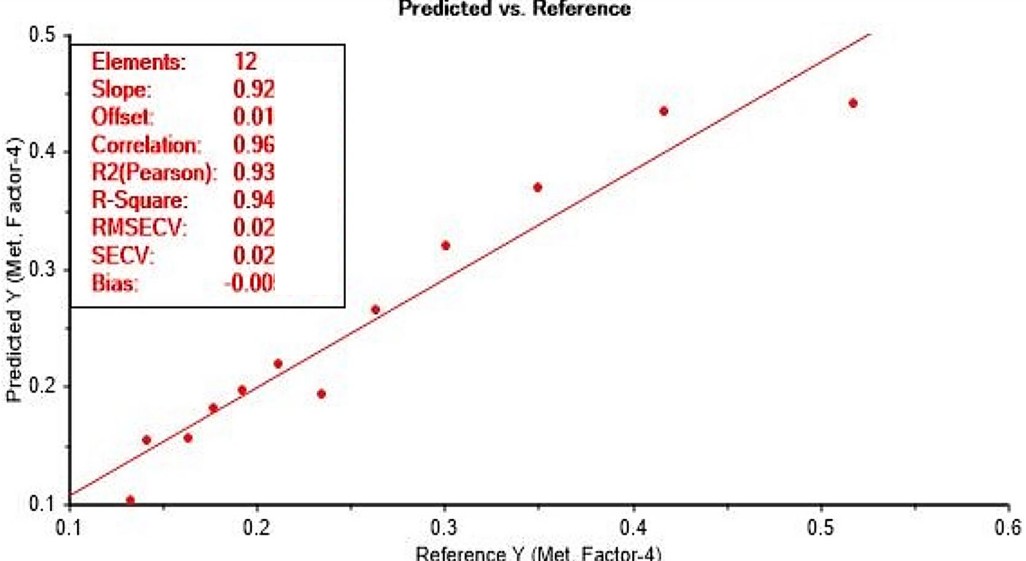

**Figure 4 Validation curve for methanol prediction using UV-Vis with acid dichromate.** Validation set for the calibration with UV-Vis (900 to 1,100 cm$^{-1}$) using acid dichromate.

for quality control of methanol in alcohol beverage. The use of acid dichromate showed strong correlation with methanol concentration in waste water from biodiesel washing observed by *Santos et al. (2023)*.

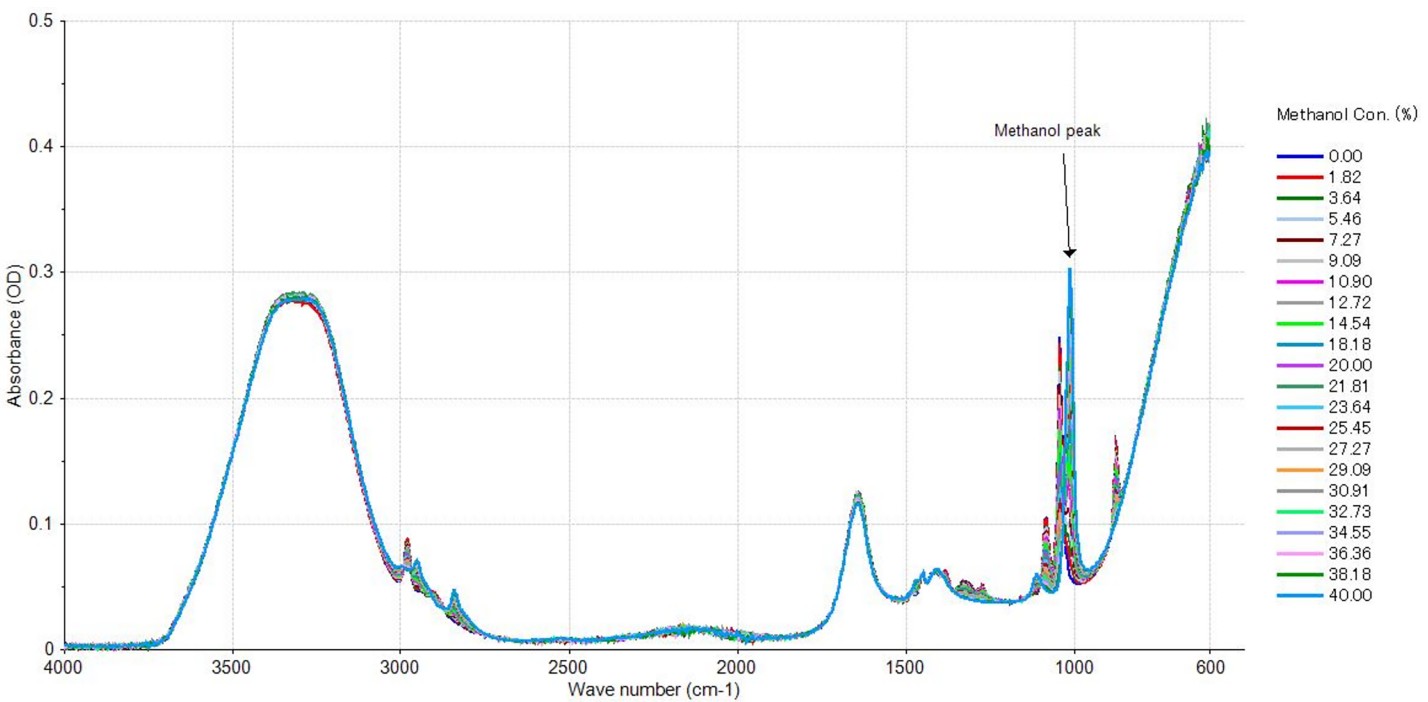

**Figure 5 Spectral distribution of spiked samples used in the methanol predictive model development.** FTIR spectral distribution for the inverse ethanol/methanol relationship. Dark blue spectra colour represents ethanol standard without methanol and light blue spectra colour represent 40% methanol.

## Identification of methanol and ethanol in the FTIR spectra of pure standards and a spiked sample

Use of IR spectroscopy for identification of alcohol species in sample mixtures is common although the methods presented challenges with accuracy and precision compared to standard methods (*Bureau, Cozzolino & Clark, 2019*; *Fakayode et al., 2020*; *Fonseca et al., 2020*). The use of NIR and MIR for quantification of ethanol in alcohol-based gel hand sanitiser during COVID 19 was possible (*Fonseca et al., 2020*), but the limit of detection was above the required standard regulated by international organisations (*Paine & Dayan, 2001*; *Botelho et al., 2020*). In order to build the FTIR model, spike volume of methanol at the expense of ethanol with DO water held constant, the pattern recognition was developed and calibrated according to spike concentration of sample mixtures. The calibration spectra were obtained at spectra region between 1,010 to 1,026 cm$^{-1}$ wavelengths with frequency intensity measured in absorbance unit (Fig. 5). A strong linearity was observed for methanol peak at 1,020 cm$^{-1}$ used in building the calibration model. The methanol peak at 1,020 cm$^{-1}$ was observed for alcohol beverage using the MIR (*Fonseca et al., 2020*) and similar observation was observed in this study using FTIR spectroscopy (Fig. 5). Several studies on methanol quantification using FTIR reported LOD and LOQ lower than the low end of the current method working range (0.18%, Table 1) due to high sensitivity of the methods and instrument. In the current study,

**Table 1 Analytical and statistical parameters of the analytical curve for each of the evaluated instrumental methods by second analyst.** Analytical curve for data collected using the UV-Vis and FTIR model.

| Instruments | Working range (%) | Analytical curve equation | $r^2$ | $RSD_r$ (%) | LOD (%) | LOQ (%) |
|---|---|---|---|---|---|---|
| ATR-FTIR | 0.18–40 | y = 0.89x − 0.11 | 0.91 | 11.4 | 0.29 | 0.89 |
| UV-Vis | 0.12–1.00 | y = 0.81x + 0.76 | 0.99 | 5.77 | 0.04 | 0.12 |

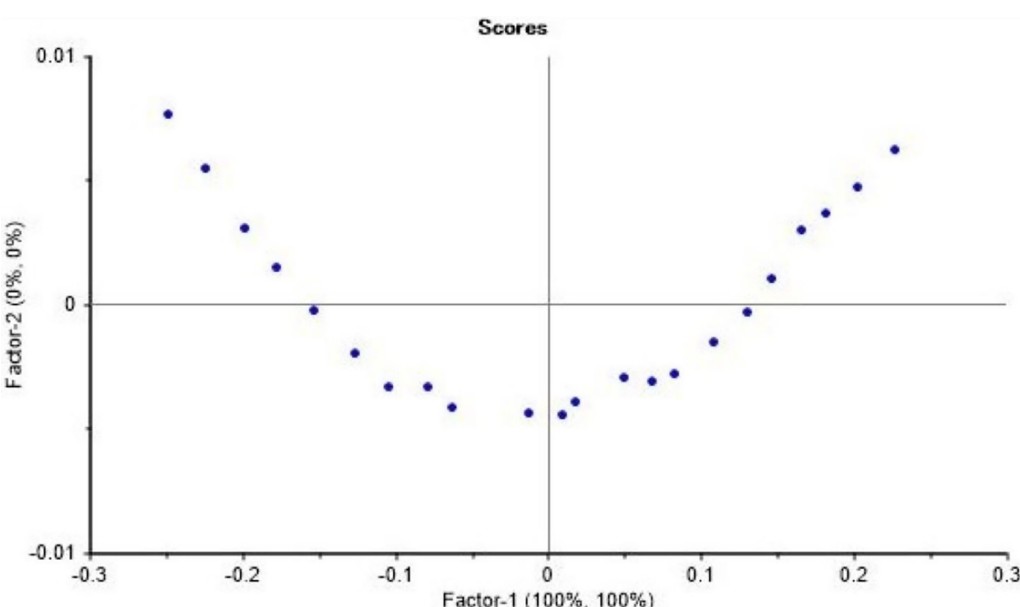

**Figure 6 Loading plot of the PLSR predictive model development.** Loading plot of the inverse relationship between ethanol and methanol used in the FTIR calibration model development.

Bruker's alpha II FTIR machine collects spectral data at every 2 nanometres by default, resulting in a low sensitivity. The FTIR used is yet useful for quality control of methanol as the LOD and LOQ are not very different from the maximum limit of methanol in alcohol beverage according to standards of various countries (*Botelho et al., 2020*). This study has confirmed the potential use of FTIR with a 2 nm interval default scan function in quality control of methanol in alcohol beverage industries.

## Calibration and validation of PLS procedure for methanol

The loading plot showed a region where methanol and ethanol concentration were equal at point 0 of the factor 1 which explain 100% of the differences (Fig. 6). The FTIR absorbance at 1,010 to 1,026 cm$^{-1}$ showed a strong correlation with methanol concentration. By applying the PLS regression for methanol calibration curve (Fig. 7 and Table 1) we were able to predict methanol content in spiked samples and test samples. Calibration model of spiked samples showed a strong linear relationship with the correlation coefficient of r = 0.997 and RMSE of 0.55% (mV$^{-1}$). The validation model also revealed a strong linear

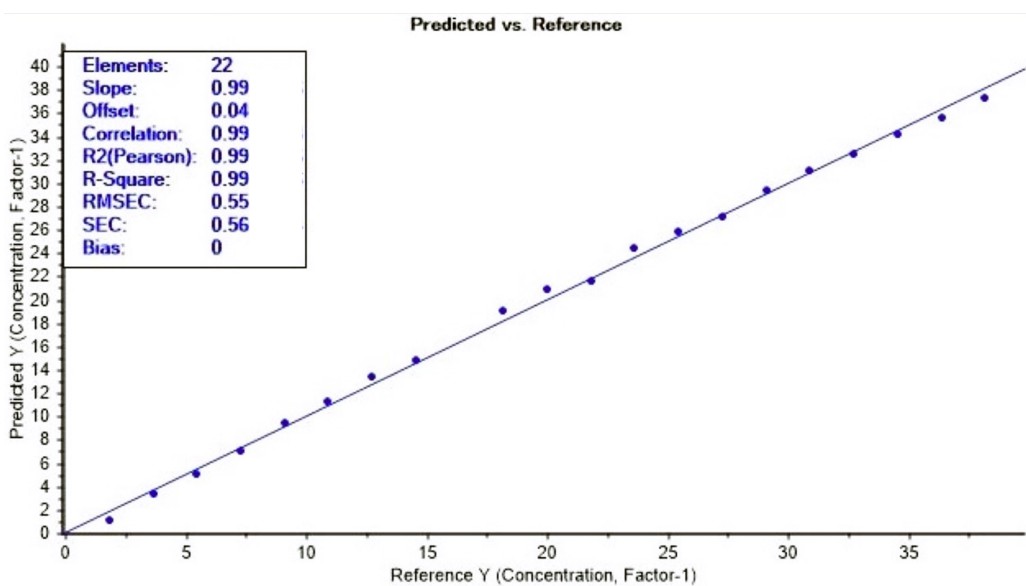

**Figure 7 Calibration curve for the FTIR calibration model for methanol prediction.** Calibration set for methanol prediction with FTIR (1,010 to 1,026 cm$^{-1}$).

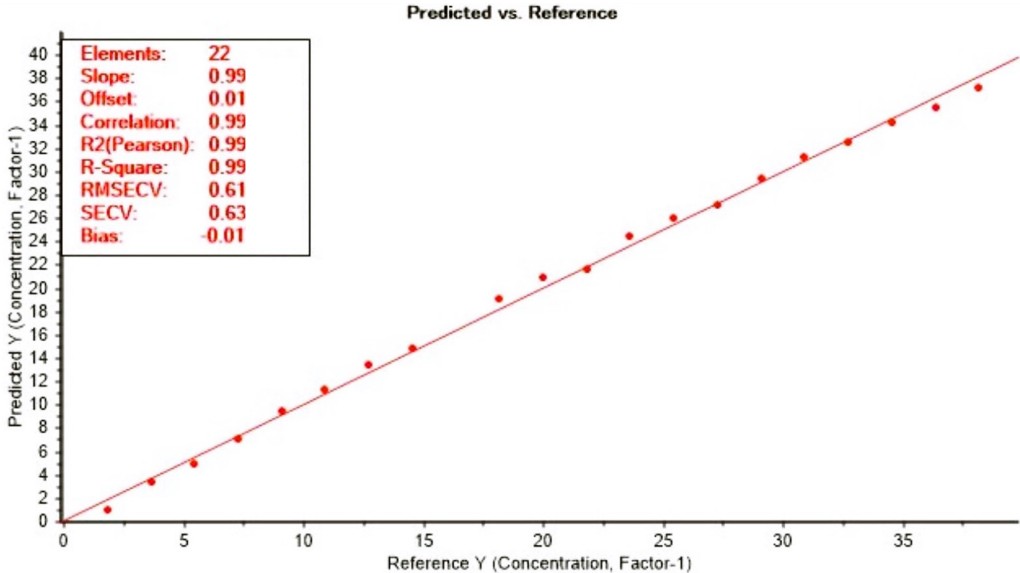

**Figure 8 Validation curve for the FTIR calibration model for methanol prediction.** Validation curve for methanol prediction with FTIR (1,010 to 1,026 cm$^{-1}$).

relationship with correlation coefficient of r = 0.998 and RMSECV of 0.62% (mV$^{-1}$) (Fig. 8). This model though was developed from a two interval spectra data point; it is still reliable for the prediction of methanol in distilled alcoholic beverages.

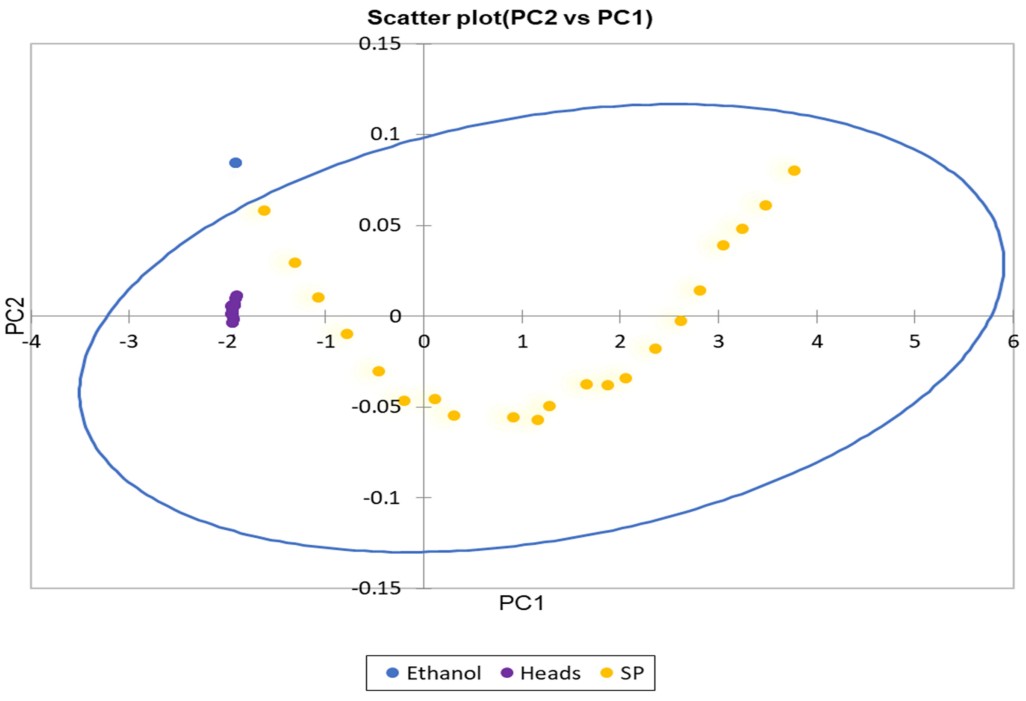

**Figure 9 The loading plot of principle component analysis of pure ethanol, spiked methanol and distilled spirit sample (Heads).** Loading plot of principle component analysis for pure ethanol, spiked (SP) methanol and distilled alcohol (Heads).

## Analytical and statistical parameters of the analytical curve for each of the evaluated instrumental methods by second analyst

The analytical curve for the FTIR method showed that limit of detection was relatively low 0.29% ($mV^{-1}$) with a correlation coefficient of r = 0.91, similar to LOD reported in previous study (*Garrigues et al., 1997*). It is worth knowing that the FTIR used in the current method collected spectral data in every two nanometre interval with an exceptional prediction model performance, making it a reliable instrument in less developed laboratory for quality control of methanol in alcohol beverages. The UV-Vis analytical curved from the current method showed low limit of detection at 0.04% ($mV^{-1}$) making it a potential method for quality control of methanol in distilled alcohol. The spectral distribution of methanol relative to ethanol in Fig. 9 revealed, most of the heads distilled alcohol are around the same concentration with spiked methanol of 0% to 1% in 40% ethanol. It was evident that the distilled alcohol (Heads) tested has methanol concentration as the distribution of samples lie within the 95% confidence eclipse (Fig. 9), however, the amount is quite low. The data generated from the prediction model revealed negative concentration and were not presented in this study, but a qualitative test using the PCA is still adequate.

The spectral distribution of 100% methanol can be easily distinguished from the rest of the spiked sample and test heads samples (Fig. 10). Figure 10 also showed the differences in

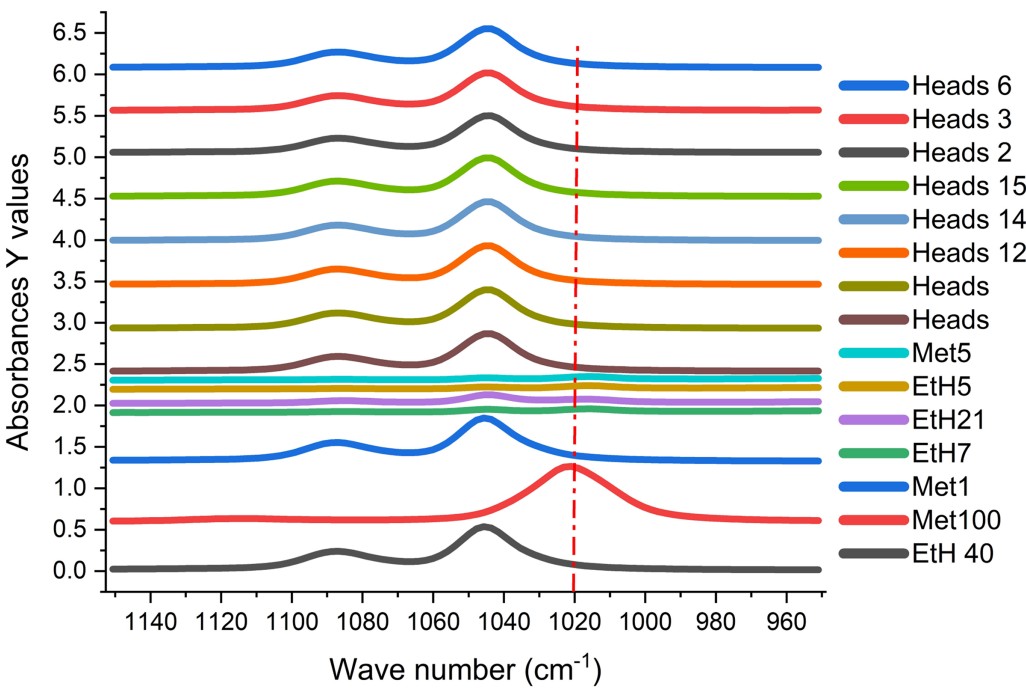

**Figure 10 Spectral distribution for methanol, ethanol, spiked standards, and distilled sample (Heads).** The FTIR spectral distribution of methanol (Met.), ethanol (EtH.) and heads distilled alcohol beverages at 950 to 1,150 cm$^{-1}$. The vertical red line (1,020 cm$^{-1}$) indicates the active peak for methanol.

spectral distribution around 1,020 cm$^{-1}$ used to build the prediction model. It is evident that quality control of methanol in distilled alcohol beverage can be easily determined using the FTIR method. A strange trend was observed in Fig. 10 for spiked methanol in ethanol where the peaks was minor as spectra were almost a straight line (Met5-Eth7), reflecting the behaviour of the mixture of spiked methanol in ethanol when concentrations are at same ratio affecting the model development.

The pure methanol solution has the characteristic vibration frequency at 1,020 cm$^{-1}$ (major signal) while ethanol had the characteristic frequency at 1,047 cm$^{-1}$ (major signal) and 1,087 cm$^{-1}$ (minor signal), similar to the observation of *Coldea et al. (2013)*. The selected absorbance of major and minor signals specific for pure methanol and ethanol clearly displayed in Fig. 10. These frequencies for methanol and ethanol are specific for stretching vibrations of C-O bonds in these molecules (*Coldea et al., 2013*).

## Implications of using UV-Vis and FTIR for quality control in the spirits industry

Using UV-Vis and FTIR in the spirits industry could enhances quality control by providing rapid, cost-effective methods for detecting impurities and ensuring product consistency. The spirits industry faces significant challenges in ensuring product quality and safety, particularly regarding the detection of harmful substances like methanol. The application of UV-Vis and FTIR spectroscopy offers promising solutions for quality

control, providing rapid, cost-effective, and reliable methods for assessing the quality of alcoholic beverages. The UV-Vis technique measures the absorbance of ultraviolet and visible light by a sample, allowing for the identification and quantification of specific compounds based on their unique absorption spectra. FTIR measures the infrared absorption of a sample, providing information about molecular vibrations and functional groups, which can be used to identify and quantify various components in a mixture. The benefits include quick analysis, significantly reducing the time required for quality control compared to traditional methods. UV-Vis and FTIR methods are cost effective, relatively inexpensive to implement, making them accessible for small and medium-sized distilleries, especially in developing regions where resources may be limited. These methods are highly sensitive and can detect low concentrations of harmful substances, such as methanol, which is crucial for consumer safety. For instance, this study demonstrated that UV-Vis could detect methanol concentrations with a limit of detection (LOD) of 0.04% $(mV^{-1})$ and FTIR with an LOD of 0.29% $(mV^{-1})$. The limit of detection (LOD) and limit of quantitation (LOQ) for FTIR method is higher than the lower working range because they represent the lowest concentration of methanol that can be reliably detected or quantified, respectively, which means they sit at the very edge of what the analytical method can accurately measure, thus falling above the lower limit of the working range where more precise measurements can be made. The regular use of these techniques can help maintain product consistency and compliance with safety regulations, thereby enhancing consumer trust and brand reputation for quality assurance. Although use of UV-Vis and FTIR are more affordable, the accuracy of results depends on proper calibration and validation of the methods, which may require initial investment in standards and training. The presence of multiple components in spirits can complicate the interpretation of spectra, necessitating advanced data analysis techniques. While these methods are gaining traction, regulatory bodies may still require traditional methods for compliance, which could limit their adoption for regulatory acceptance.

## Improving accuracy in UV-Vis and FTIR methods for methanol quantification

Quantifying methanol accurately using UV-Vis and FTIR spectroscopy is crucial in various fields, including environmental monitoring and chemical manufacturing. Several strategies could enhance the accuracy of these methods includes enhanced calibration techniques by developing robust calibration curves using a range of known methanol concentrations can significantly improve accuracy. Matrix matching is essential to ensure that the calibration samples are prepared under the same conditions as the unknown samples to minimize variability. The presence of other substances can interfere with the absorption spectra. Using matrix-matched standards can help mitigate this issue, ensuring that the calibration reflects the actual sample matrix. Use of advanced spectroscopic techniques such as derivative Spectroscopy techniques can enhance the resolution of overlapping peaks in UV-Vis spectra, allowing for better quantification of methanol in complex mixtures. Utilizing FTIR with attenuated total reflectance (ATR) can improve the sensitivity and specificity of methanol detection. This method allows for direct analysis of

liquid samples without extensive sample preparation. Incorporation of machine learning algorithm can enhance interpretation of data as well.

Implementing machine learning algorithms can enhance the interpretation of spectral data. Techniques such as partial least squares regression (PLSR) or support vector machines (SVM) can be trained on spectral data to predict methanol concentrations more accurately. Utilizing multivariate statistical methods can help in understanding the relationships between different variables in the spectral data, leading to improved quantification accuracy. Sample preparation and handling is also a challenge that needs to be addressed for consistency. Ensuring that samples are prepared consistently (*e.g.*, same dilution factors, temperature, and handling procedures) can reduce variability in results. Additionally, minimizing contamination by using clean glassware and avoiding cross-contamination during sample preparation are critical for maintaining the integrity of the samples. Regular maintenance and calibration of UV-Vis and FTIR instruments are essential to ensure they operate within specified parameters, thus improving measurement accuracy. Finally, inclusion of quality control samples in each batch of analyses can help monitor the performance of the analytical methods and identify any deviations from expected results. By implementing these strategies, the accuracy of UV-Vis and FTIR methods for methanol quantification can be significantly improved, leading to more reliable and reproducible results.

## CONCLUSION

UV-Vis and FTIR method can be useful not only for determining the methanol and ethanol content in spirits, but also to evaluate the ratio between these two volatile components. UV-Vis and FTIR PLS offers valuable advantages when choosing *vs* conventional methods such as GC-MS. Their importance is justified by being a rapid, efficient and non-destructive tool for screening alcoholic beverages. Besides, quantitative PLS regression analysis with FTIR spectral data is more practical in developing countries as high end instruments are not affordable. The UV-Vis acid dichromate method can be utilised also with simple linear regression for peak absorbance at 975 cm$^{-1}$ spectra region. These two method becomes important for authenticity control based on provenience region for desired distilled alcohol beverages. This study has confirmed the utilization of the two methods in quantification and qualification of distilled alcohol beverage for high alcohol.

## ACKNOWLEDGEMENTS

We acknowledge the support of Dr. Adel B. from the CEFAS UK for providing the instruments needed for this study.

### Funding

The authors received no funding for this work.

## Competing Interests

The authors declare that they have no competing interests.

## Author Contributions

- Ronick S. Shadrack conceived and designed the experiments, performed the experiments, analyzed the data, performed the computation work, prepared figures and/or tables, authored or reviewed drafts of the article, and approved the final draft.
- Krishna K. Kotra conceived and designed the experiments, performed the experiments, analyzed the data, prepared figures and/or tables, authored or reviewed drafts of the article, and approved the final draft.
- Daniel Tari performed the experiments, analyzed the data, authored or reviewed drafts of the article, and approved the final draft.
- Hancy Tabi performed the experiments, analyzed the data, authored or reviewed drafts of the article, and approved the final draft.
- Jacinta Botleng performed the experiments, analyzed the data, authored or reviewed drafts of the article, and approved the final draft.
- Rolina Kelep performed the experiments, analyzed the data, authored or reviewed drafts of the article, and approved the final draft.
- Ladyshia Regenvanu performed the experiments, analyzed the data, authored or reviewed drafts of the article, and approved the final draft.

## Data Availability

The spectral data for the two methods are in the Supplemental File.

## Supplemental Information

Supplemental information for this article can be found online at http://dx.doi.org/10.7717/peerj-achem.35#supplemental-information.

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
