# Peer review of "A rapid method for methanol quantification in spirits using UV-visible spectroscopy and FTIR"

_PeerJ Analytical Chemistry, doi:10.7717/peerj-achem.35_

## Round 0.1 · original submission · Major Revisions

The reviewers raised a number of points that need to be taken into account by the authors while revising the manuscript.

·

Basic reporting

1. This manuscript should be submitted to a professional research paper writing service to improve its language quality and address any linguistic inaccuracies.
Examples:
I. The term "ultraviolet-visible spectrometry" is abbreviated as both "UV Vis" and "UV VIS" in various sections. To enhance clarity for readers, it is recommended to adhere to ISO standards and maintain a consistent abbreviation throughout the article.
II. There are capitalization errors in the text, such as in line 31 where "Methanol" is capitalized in the middle of a sentence. Chemical names should not be capitalized unless they appear at the beginning of a sentence.
2. In the introduction section, references must be provided for many sentences. However, widely recognized public information does not require citation. Any sentence that is based on another person's work must be properly cited.
Example:
I. Line 56. “Although IR spectroscopy has some advantages over UV VIS in terms of wide range of sample types, they are both important in food industry for quality control measures.”
3. Figures and quality:
I. Some of the figures appear pixelated. It would be beneficial to improve their resolution to enhance the reading experience for the audience.
II. Please ensure that all values in the figures are rounded to three significant digits.
III. In Figure 2 (UV-Vis spectra 1), the Y-axis (absorbance) should start from zero. Additionally, please reduce the number of ticks on the wavenumber axis and increase the font size for better graph visibility.
IV. In Figure 5 (FTIR spectra 1), it would be helpful to label the methanol peak.
V. In Figure 9, kindly remove the labels on the data points and consider either increasing the size of the dots or using different shapes for ethanol, heads, and SP. Increasing the font size will also improve readability.
VI. Figure 10 looks good in terms of resolution; no changes are needed for this figure.

Experimental design

The current manuscript is appropriate for the aims and scope of the PeerJ Analytical Chemistry journal.
2. The methods involving UV-Vis and FTIR are described in sufficient detail.
3. In line 153, the author states, "As shown in Equation 1," however, Equation 1 is not provided.

Validity of the findings

1. The manuscript titled “A Rapid Method for Methanol Quantification in Spirits Using UV-Visible Spectroscopy and FTIR” (#104989) presents techniques for determining methanol content in spirits through the use of UV-Vis and FTIR spectroscopy. The UV-Visible spectrometry method described involves the use of potassium dichromate in acidic conditions to oxidize alcohols, leading to the reduction of chromium from the Cr (VI) (yellow color) oxidation state to Cr (III) (green color). It is worth noting that this method has been widely recognized since the 1990s. (Hari, M.; Deoki, N. Spectrophotometric Determination of Some Monohydric Alcohols Based on Their Oxidation by K2CrO4-HNO3 System. Indian J. Chem 1994, 33, 359–361. ; Tgd, S.; Rsrd, S.; Fgd, C.; Dda, S. ATR-FTIR and UV-Vis as Techniques for Methanol Analysis in Biodiesel-Washing Wastewater. QuÌmica Nova 2023, 46, 698–705.). Previously published methods have reported alcohols (ethanol and methanol) at a wavenumber of 600 cm⁻¹. The current manuscript provides a clarification regarding the correlation of methanol at 970 cm⁻¹, which is considered one of the novel aspects of this study. To enhance clarity, it would be beneficial to highlight this correlation by zooming in on the 970 cm⁻¹ region in Figure 2.

2. Another commonly reported method, FTIR, is widely recognized for its effectiveness in quantifying ethanol and methanol content in various samples (Pérez-Ponce, A.; de la Guardia, M. Partial Least-Squares–Fourier Transform Infrared Spectrometric Determination of Methanol and Ethanol by Vapour-Phase Generation. Analyst 1998, 123, 1253–1258. ; Coldea, T. E.; Socaciu, C.; Fetea, F.; Ranga, F.; Pop, R. M.; Florea, M. Rapid Quantitative Analysis of Ethanol and Prediction of Methanol Content in Traditional Fruit Brandies from Romania, Using FTIR Spectroscopy and Chemometrics. Not. Bot. Horti Agrobot. Cluj Napoca 2013, 41, 143.). Could you kindly compare the previously established FTIR methods for ethanol and methanol quantification with the current method you have developed, and emphasize the significant improvements? It is important to clearly demonstrate the advancements made, as presenting a method without novelty or improvement may not add substantial value to the literature.

Reviewer 2 ·

Basic reporting

The English language is good and acceptable.

Literature review is not sufficient. Following related articles should be reviewed and discussed and should be cited in the Introduction:

Debebe et al., Alcohol determination in distilled alcoholic beverages by liquid phase Fourier transform mid-infrared and near-infrared spectrophotometries. Food Analytical Methods, 2017, 10, 172-179.
Debebe et al., Non-destructive determination of ethanol levels in fermented alcoholic beverages using Fourier transform mid-infrared spectrometry, Chemistry Central Journal, 2017, 11, 27. DOI 10.1186/s13065-017-0257-5.
Debebe et al., Partial least square-ultra violet near infrared spectrometric determination of ethanol in distilled alcoholic beverages and its comparison with reference method, gas chromatography. Bulletin of the Chemical Society of Ethiopia, 2017, 31, 201-209.

Experimental design

The experimental design is good. Research question is well defined, relevant and meaningful.

Methods used have been described with sufficient detail.

Validity of the findings

The described method has been validated.

Conclusions are well stated.

Additional comments

Line 28, ration should be ratio.

Line 31, Methanol, no need to use capital letter in the middle of sentences.

Line 38, give working range before giving LOD and LOQ

Line 50, Introduction is bit short. Literature review is not comprehensive. Following related relevant articles should be reviewed and discussed and cited in the Introduction:

Debebe et al., Alcohol determination in distilled alcoholic beverages by liquid phase Fourier transform mid-infrared and near-infrared spectrophotometries. Food Analytical Methods, 2017, 10, 172-179.
Debebe et al., Non-destructive determination of ethanol levels in fermented alcoholic beverages using Fourier transform mid-infrared spectrometry, Chemistry Central Journal, 2017, 11, 27. DOI 10.1186/s13065-017-0257-5.
Debebe et al., Partial least square-ultra violet near infrared spectrometric determination of ethanol in distilled alcoholic beverages and its comparison with reference method, gas chromatography. Bulletin of the Chemical Society of Ethiopia, 2017, 31, 201-209.

Line 85, SI unit for volume is “L” therefore, replace “l” by “L”

Table 1,
Instruments Working range (%) RSDr (%) LOD (%) LOQ (%)
ATR-FTIR 0.182-40 11.4 0.29 0.89
UV-Vis 0.125-1 15 0.50 1.50

How can LOD and LOQ much higher than the lower end of working range? This is against the analytical practice? The author should give convincing explanation for this?

---

## Round 0.2 · Minor Revisions

Reviewer 1 has some suggestions for improving the manuscript.

·

Basic reporting

In line 71, the "S" in "Spectroscopy" should be lowercase.

Please review the capitalization of letters in the sentences carefully.

In line 95, "FDA" is incorrectly abbreviated as the "Food and Drugs Association of America." It should be correctly abbreviated as "Food and Drug Administration" or "U.S. Food and Drug Administration."

In the caption of Figure 8, the unit "cm⁻¹" in "FTIR (1010 cm-1 to 1026 cm-1)" should be properly superscripted.

Experimental design

No comment

Validity of the findings

No comment

Reviewer 2 ·

Basic reporting

Clear and unambiguous, professional English used throughout.

Literature references, sufficient field background/context provided.

Professional article structure, figures, tables. Raw data shared.

Self-contained with relevant results to hypotheses.

Experimental design

Original primary research within Aims and Scope of the journal.

Research question well defined, relevant & meaningful. It is stated how research fills an identified knowledge gap.

Rigorous investigation performed to a high technical & ethical standard.

Methods described with sufficient detail & information to replicate.

Validity of the findings

Meaningful replication encouraged where rationale & benefit to literature is clearly stated.

All underlying data have been provided; they are robust, statistically sound, & controlled.

Conclusions are well stated, linked to original research question & limited to supporting results.

Additional comments

The manuscript has been revised appropriately. The revised manuscript is acceptable.

---

## Round 0.3 · accepted · Accept

The reviewers support publication of this revised manuscript.

·

Basic reporting

no comment

Experimental design

no comment

Validity of the findings

no comment

Additional comments

no comment